# Alpha Net: Adaptation with Composition in Classifier Space

## Abstract

Deep learning classification models typically train poorly on classes with small numbers of examples. Motivated by the human ability to solve this task, models have been developed that transfer knowledge from classes with many examples to learn classes with few examples. Critically, the majority of these models transfer knowledge within *model feature space*. In this work, we demonstrate that transferring knowledge within *classifier space* is more effective and efficient. Specifically, by linearly combining strong nearest neighbor classifiers along with a weak classifier, we are able to compose a stronger classifier. Uniquely, our model can be implemented on top of any existing classification model that includes a classifier layer. We showcase the success of our approach in the task of long-tailed recognition, whereby the classes with few examples, otherwise known as the "tail" classes, suffer the most in performance and are the most challenging classes to learn. Using classifier-level knowledge transfer, we are able to drastically improve - by a margin as high as 10.5% - the state-of-the-art performance on the "tail" categories.

## 1 Introduction

The computer vision field has made rapid progress in the area of object recognition due to several factors: complex architectures, larger compute power, more data, and better learning strategies. However, the standard method to train recognition models on new classes still relies on training using large sets of examples. This dependence on large scale data has made learning from few samples a natural challenge. Highlighting this point, new tasks such as low-shot learning and long-tailed learning, have recently become common within computer vision.

Many approaches to learning from small numbers of examples are inspired by human learning. In particular, humans are able to learn new concepts quickly and efficiently over only a few samples. The overarching theory is that humans are able to transfer their knowledge from previous experiences to bootstrap their new learning task (Lake et al., 2017; 2015; Gopnik & Sobel, 2000).

Inherent in these remarkable capabilities are two related questions: what knowledge is being transferred and how is this knowledge being transferred? Within computer vision, recent low-shot learning and long-tailed recognition models answer these questions by treating visual "representations" as the knowledge structures that are being transferred. As such, the knowledge transfer methods implemented in these models transfer learned features from known classes learned from large data to the learning of new classes with low data (Liu et al., 2019; Yin et al., 2019). These models exemplify the broader assumption that, in both human and computer vision, knowledge transfer occurs within model representation and feature space (Lake et al., 2015). In contrast, we claim that previously learned information is more concisely captured in classifier space. This inference is based on the fact that sample representation is unique to that sample, but classifiers are fitted for all the samples in a given class. The success of working within classifier space to improve certain classifiers has been established in several papers (Elhoseiny et al., 2013; Qi et al., 2018), where the models are able to directly predict classifiers from features or create new models entirely by learning other models. Other non-deep learning models use classifiers learnt with abundant data to generate novel classifiers (Aytar & Zisserman, 2011; 2012).

Despite these successes, the concept of learning within classifier space is not as common in deep learning models. We suggest that transfer learning can, likewise, be implemented in the classifier space of deep learning models. Specifically, we combine known, strong classifiers (*i.e.*, learned with large datasets) with weak classifiers (*i.e.*, learned with small datasets) to improve our weak classifiers.

Our classifier space method is illustrated in Figure 1. In this toy example demonstrating our method, we are given $n$ classifiers $C_i$ trained with large data

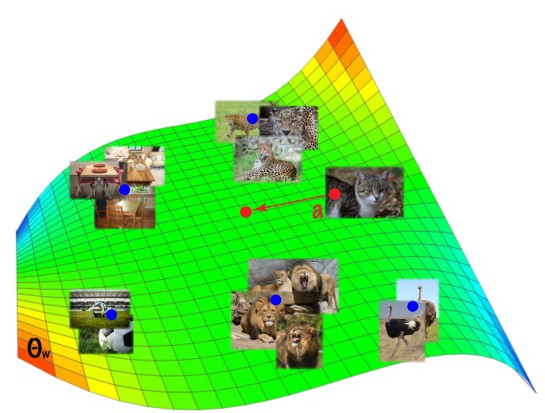

Figure 1: A classifier space depicting how Alpha Net adaptively adjusts weak classifiers through nearest neighbor compositions.

and a weak classifier $a$, which was trained for a class with very few samples. Our goal is to combine the most relevant strong classifiers to adaptively adjust and improve $a$. Our method implements this approach in the simplest way possible. To select the most effective strong classifiers to combine, we take the target's $a$ nearest neighbor classifiers. Given the appropriate nearest neighbor classifiers, the challenge becomes how to combine these strong classifiers with the target weak classifier so as to improve its performance given no further samples.

We address this challenge using what we view as the most natural approach - creating a new classifier by *linearly* combining the nearest neighbor classifiers and the original weak classifier. Embedded in this solution is the further challenge of choosing a strategy for computing the combination coefficients. We propose learning the coefficients of the linear combination through another neural network, which we refer to as "Alpha Net". As compared to many other approaches to learning from small numbers of examples, our methodology has three characteristics:

1. Our approach can be implemented on top of any architecture. This is because the Alpha Net does not need to re-learn representations and only operates within classifier space to improve weak classifiers. As a result, our approach is agnostic to the type of architecture used to learn the classifiers; it merely provides a systematic method for combining classifiers.

2. Our approach demonstrates the importance of combining not only the most relevant classifiers but also the original classifier. In the absence of the original classifier, any combination of classifiers becomes a possible solution, without being constrained by the initial classifier.

3. Our approach creates for every target class a completely different set of linear coefficients for our new classifier composition. In this manner, we are learning our coefficients in a more *adaptive* way, which is extremely difficult to achieve through classical methods.

To illustrate the efficacy of our method, we apply it to the task of long-tailed recognition. "Long-tailed" refers to a data distribution in which there is a realistic distribution of classes with many examples (head classes) and classes with few examples (tail classes). We compare our Alpha Net method to recent state-of-the-art models Kang et al. (2020) on two long-tailed datasets: ImageNet-LT and Places-LT. Critically, we are able to improve the tail classifiers accuracy by as much as 10.5%.

## 2 RELATED WORK

Creating, modifying, and learning model weights are concepts that are seen in many earlier models. In particular, these concepts appear frequently in transfer learning, meta-learning, low-shot learning, and long-tailed learning.

**Classifier Creation** The process of creating new classifiers is captured within meta-learning concepts such as learning-to-learn, transfer learning, and multi-task learning (Thrun & Pratt, 2012;

Schmidhuber et al., 1997; Pan & Yang, 2009; Caruana, 1997; Santoro et al., 2016). These approaches generalize to novel tasks by learning shared information from a set of related tasks. Many studies find that shared information is embedded within model weights, and, thus, aim to learn structure within learned models to directly modify another network's weights (Schmidhuber, 1987; 1992; 1993; Bertinetto et al., 2016; Ha et al., 2016; Finn et al., 2017; Rebuffi et al., 2017; Sinha et al., 2017; Munkhdalai & Yu, 2017). Other studies go even further and instead of modifying networks, they create entirely new networks from training samples only (Socher et al., 2013; Lei Ba et al., 2015; Noh et al., 2016). However, our method only combines existing classifiers, without having to create new classifiers or networks from scratch.

**Classifier or Feature Composition** In various classical approaches, there has been work that learns better embedding spaces for image annotation (Weston et al., 2010) or uses classification scores as useful features (Wang et al., 2009). However, they do not attempt to compose classifiers nor do they address the long-tail problem. Within non-deep methods in classic transfer learning, there have been attempts of using and combining SVMs. In Tsochantaridis et al. (2005), SVMs are trained per object instance and a hierarchical structure is required for combination in the datasets of interest. Such a structure is typically not guaranteed nor provided in our long-tail datasets. Additional SVM work uses regularized minimization to learn the coefficients necessary to combine patches from other classifiers (Aytar & Zisserman, 2012). While conceptually similar to Alpha Net, our method has an additional advantage of learning the compositional coefficients in a more adaptive way. Specifically, different novel classes will have their own sets of alphas, and similar novel classes will naturally have similar sets of alphas. Learning such varying sets of alphas is difficult in previous classical approaches. They either learn a fixed set of alphas for all of the novel classes or are forced to introduce more complex group sparsity-like constraints. Finally, in zero-shot learning there exist methods which compose classifiers of known visual concepts to learn a completely new classifier (Elhoseiny et al., 2013; Misra et al., 2017; Lei Ba et al., 2015; Changpinyo et al., 2016). However, such composition is often guided by additional attribute supervision or textual description, which Alpha Net does not depend on.

**Learning Transformations Between Models and Classes** Other studies have demonstrated different ways of learning transformations to modify model weights in an attempt to learn these transformations during stochastic gradient descent (SGD) optimization (Andrychowicz et al., 2016; Ravi & Larochelle, 2017). Additionally, Wang & Hebert (2016) empirically show the existence of a generic nonlinear transformation from small-sample to large-sample models for different types of feature spaces and classifier models. Finally, Du et al. (2017) provide theoretical guarantees on performance when one learns the transformation going from the source function to a related target function. In our work, we are also inferring that our target classifier is a transformation from a set of source classifiers.

**Low-shot Learning** Meta-learning, transfer learning, and learning-to-learn are frequently applied to the domain of low-shot learning (Fei-Fei et al., 2006; Koch et al., 2015; Lake et al., 2015; Santoro et al., 2016; Wang & Hebert, 2016; Li & Hoiem, 2017; Hariharan & Girshick, 2017; Bromley et al., 1994; Koch et al., 2015; Snell et al., 2017; George et al., 2017; Wang et al., 2017; Akata et al., 2015). Many prior studies have attempted to transfer knowledge from tasks with abundant data to completely novel tasks (Vinyals et al., 2016; Ravi & Larochelle, 2017; Bertinetto et al., 2016). However, the explicit nature of low-shot learning consisting of tasks with small fixed samples means that these approaches do not generalize well beyond the arbitrary few tasks. This is indeed a significant problem as the visual world very clearly involves a wide set of tasks with continuously varying amounts of information.

**Long-tailed Learning** The restrictions on low-shot learning have directly led to a new paradigm referred to as "long-tailed" learning, where data samples are continuously decreasing and the data distribution closely models that of the visual world. Recent work has approached long-tailed recognition by re-balancing the samples in different stages of model training (Cao et al., 2019). One other study attempts to transfer features from common classes to rare classes (Liu et al., 2019) or transfer intra-class variance (Yin et al., 2019). However, both these approaches to knowledge transfer require complex architectures, such as a specialized attention mechanism and memory models as in Liu et al. (2019). While most studies have largely focused on representation space transferability, recent work explores the potential of operating in classifier space (Kang et al., 2020). Results suggest that decoupling model representation learning and classifier learning is actually a more efficient

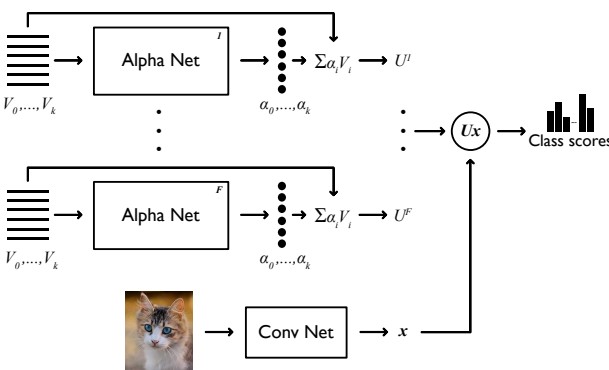

Figure 2: An illustration of Alpha Net. "Conv Net" is a stand-in for any fixed pre-trained model with a classification layer. For figure simplicity, the bias is omitted; $V_k^i$ and $\alpha_k^i$ are denoted as $V_k$ and $\alpha_i$.

way to approach long-tailed learning. Specifically, normalizing classifiers and adjusting classifiers only with re-sampling strategies are able to achieve good performance. Their success in working only with classifiers supports our general concept that combining strong classifiers is a natural and direct way to improve upon weak classifiers.

## 3 METHODS

Long-tailed recognition is uniquely characterized by its unbalanced, long-tailed distribution of data. Due to the lack of sufficient data in parts of the distribution, otherwise known as the "tail", training on the tail classes significantly suffers compared to that of "base" classes that have sufficient data. As a result, tail classes typically see low accuracy performance, and this remains the dominating challenge of long-tailed recognition. Our method directly addresses this issue, and successfully improves the tail performance by combining classifiers in order to adaptively adjust weak classifiers.

### 3.1 NOTATION

We first split up our long tail dataset into two broad splits: our "base" classes with $B$ number of classes and our "few" $F$ classes. The "base" classes contain classes with many examples, and conversely the "few" classes contain classes with few examples. We denote by $N = B + F$ the total number of classes.

Additionally, we denote a generic sample by $I$ and its corresponding label by $y$. We split the training set into two subsets of pairs $(I_i, y_i)$: the subset $X_{base}$ contains samples from the $B$ classes, and the subset $X_{few}$ contains samples from the $F$ classes.

Finally, we work with a fixed pre-trained model with classifiers $W_j$ and biases $b_j$, $j \in (1, \ldots, N)$; here, classifiers are the last layers of networks, which map input representations to class scores. Given target class $j$, and its classifier $W_j$ and bias $b_j$, let class $j$'s top $k$ nearest neighbor classifiers and biases be defined as $V_i^j$ and $p_i^j$, $i \in (1, \ldots, K)$. Furthermore, we also define the trivial case where the immediate nearest neighbor is itself, $V_0^j = W_j$ and $p_0^j = b_j$.

### 3.2 ALPHA NET

Our combination model, Alpha Net, takes in trained classifiers from the $B$ "base" classes and weak classifiers from the $F$ "few" classes, in order to learn a new classifier composition for each "few" class. The following section will detail the model and our training as depicted in Figure 2.

**Architecture** In order to create effective individual classifiers, Alpha Net is split up into several sub-modules, $A_i$, $i \in (1, \ldots, F)$. Each $A_i$ is responsible for the composition of a new classifier for class $i$, and consists of 2 fully connected layers with a ReLU in between. Its input is a flattened concatenated vector of $V_k^i$, $k \in (0, \ldots, K)$, where $V_0^i = W_i$, the original classifier for class $i$. Its output is an $\alpha^i$ vector, where $\alpha_k^i$, $k \in (0, \ldots, K)$ corresponds to input $V_k^i$.

**Training** Alpha Net is trained with all $X_{few}$ and an equal number of random samples from $X_{base}$ acting as negative samples. For each $A_i$, its input, $V_k^i$, $k \in (0, \ldots, K)$ is fixed for the entirety of training. After the inputs are forward through $A_i$, we clamp $\alpha_0^i$ (which is responsible for $V_0^i$), such that $0 < \alpha_0^i \leq \gamma$, a hyper-parameter.

After clamping, given our $\alpha_k^i$ and input $V_k^i$, $k \in (0, \ldots, K)$, we define our new classifier and bias for class $i$ as

$$U_i = \sum_{j=0}^{K} \alpha_j^i \cdot V_j^i. \tag{1}$$

$$t_i = \sum_{j=0}^{K} \alpha_j^i \cdot p_j^i. \tag{2}$$

For each $A_i$ we sequentially perform clamping, Eq. 1, and Eq. 2 only once to learn $U_i$ and $t_i, i \in (1, \ldots, F)$ for each target class.

It is important to note that while we are attempting to compose new "few" classifiers, we are still working on a long-tailed recognition problem. Thus, when training we must also consider our "base" classifiers that have been sufficient trained. Now, given a training image sample and label pair $\{I_i, y_i\}$ and its final feature representation $x_i$, we compute our sample score as

$$s_{if} = x_i \cdot U_f + t_f. \qquad f \in (1, \ldots, F) \tag{3}$$

$$s_{ib} = x_i \cdot W_b + t_b. \qquad b \in (1, \ldots, B) \tag{4}$$

Finally, we compute softmax cross entropy loss on *s* and backpropagate throughout Alpha Net and its modules.

## 4    EXPERIMENTS

### 4.1    EXPERIMENT SETUP

**Datasets** To properly compare our approach to state-of-the-art models as discussed in Kang et al. (2020), we perform all of our experiments using the same two long-tailed benchmark datasets: ImageNet-LT and Places-LT (Liu et al., 2019). Both datasets are sampled from their respective original datasets, ImageNet (Russakovsky et al., 2015) and Places365 (Zhou et al., 2017), such that the new distributions follow a standard long-tailed distribution. ImageNet-LT contains 1000 classes with the number of samples per class ranging from 5 to 4980 images. Places-LT contains 365 classes with the number of samples per class ranging from 5 to 1280 images.

ImageNet-LT and Places-LT are originally broken down into three broad splits that indicate the number of samples per class: 1) "many" contains classes with greater than 100 samples; 2) "medium" contains classes with greater than or equal to 20 samples but less than or equal to 100 samples; 3) "few" contains classes with less than 20 samples. In our experiments we typically combine the "many" and "medium" splits together and refer to this set of classes as the "base" split.

**Network and Data Setup** Alpha Net is designed such that there is an individual alpha sub-module for every target class for which we will recompose its weak classifier. In our experiments, we only adjust the weak classifiers belonging to the "few" split of each dataset. Therefore, we have exactly $F$ number of sub-modules for all of our Alpha Nets. Due to the number of sub-modules, as well as to facilitate more efficient computation, we reduce the dimensionality of our input classifiers through PCA. For ImageNet-LT, all classifiers are reduced from 2048 dimensions to 500. Because there are a total of 365 classes in Places-LT, all classifiers are reduced from 2048 dimensions to 300. Finally, for alpha sub-modules, we set the dimension of our 2$^{nd}$ fc layer to be equal to our reduced classifier

| Methods | Few | | Medium | | Many | | All | |
|---|---|---|---|---|---|---|---|---|
| | top1 | top5 | top1 | top5 | top1 | top5 | top1 | top5 |
| Joint† | 7.7 | - | 37.5 | - | 65.9 | - | 44.4 | - |
| NCM† | 28.1 | - | 45.3 | - | 56.6 | - | 47.3 | - |
| t-normalized† | 30.7 | - | 46.9 | - | 59.1 | - | 49.4 | - |
| cRT⋆ | 27.4 | 57.3 | 46.2 | 73.4 | 61.8 | 81.8 | 49.6 | 74.4 |
| LWS⋆ | 30.3 | 61.5 | 47.2 | 73.7 | 60.2 | 81.6 | 49.9 | 75.1 |
| Alpha - cRT | **34.5** | **63.1** | 44.3 | 72.5 | 60.5 | 81.1 | 49.2 | 74.6 |
| Alpha - LWS | **40.8** | **67.6** | 43.2 | 72.2 | 57.6 | 80.6 | 48.4 | 74.8 |

Table 1: ImageNet-LT top1 and top5 accuracy for Alpha Net. Methods with † indicate that the results were directly taken from Kang et al. (2020). Methods with ⋆ indicate that the results were reproduced and verified to be the same as that reported in Kang et al. (2020).

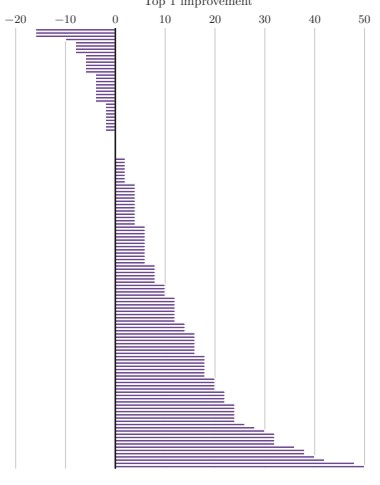

(a) Histogram of classwise improvement for Alpha cRT (best few).

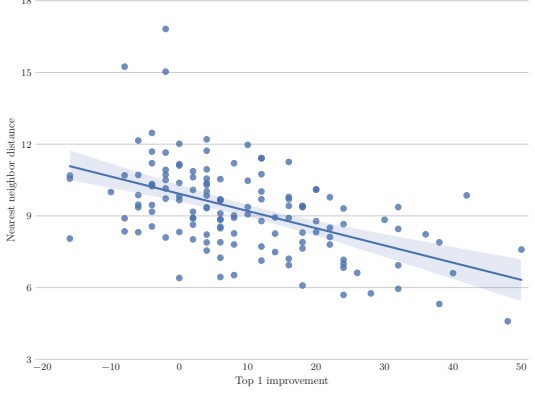

(b) A depiction of the relationship between classwise improvement and the l2 distance to its closest neighbor for Alpha cRT (best few).

Figure 3: Class improvements in average top 1 accuracy for classes in "few" test set of ImageNet-LT.

dimension size. That is, for ImageNet-LT Alpha Net, the 2$^{nd}$ fc layer is of dimension 500; for Places-LT Alpha Net, the 2$^{nd}$ fc layer is of dimension 300.

**Implementation** All experiments were run using the PyTorch framework. To be comparable to Kang et al. (2020), we perform most of our experiments using the ResNeXt-50 backbone. Furthermore, as our method is designed to run on top of pre-trained models, we use models from Kang et al. (2020) in order to be directly comparable to their results. All models used for training were downloaded directly from the GitHub repository of Kang et al. (2020). Unless specified otherwise, all experiments are trained with SGD optimizer of 0.9 momentum, learning rate 0.1, batch size 64, weight decay by an order of 0.1 every 20 epochs, and image size of $224 \times 224$. All Alpha Nets were trained for 80 epochs. All models reported are selected by cross-validation.

### 4.2 RESULTS AND ANALYSIS

**State-of-the-Art Comparison**

We ran Alpha Net using state-of-the-art models on ImageNet-LT and showcase our improvement. According to Kang et al. (2020), the best two models are: 1) Model with a re-trained classifier

on class-balanced samples, denoted as classifier re-training (cRT); 2) Model trained with a learnt scalar for weight normalization, denoted as learnable weight scaling (LWS). We directly run Alpha Net on the classifiers taken from their publicly available ResNeXt-50 models on cRT and LWS (which we refer to as Alpha cRT and Alpha LWS) and evaluate with the corresponding fixed models' convolutional weights. Alpha cRT and Alpha LWS have an alpha sub-module for every class in the "few" split. Our inputs to each alpha sub-module are the top 5 nearest neighbor classifiers with the addition of the original weak classifier for the target class - leading to a total of 6 classifiers as input. For each target class, we take its mean feature from the last layer before "fc" and select nearest neighbor classes from "base" classes that have the most similar class mean features in euclidean space.

In Table 1, we present our ImageNet models with the best gamma selected ($\gamma = 0.7$) through cross-validation. As seen in Table 1, our models maintain overall state-of-the-art cRT and LWS performance, while outperforming cRT and LWS by 7.1 and 10.5 points respectively in the "few" split, the hardest challenge in long-tail recognition. Concisely, we are able to significantly improve on "few" with comparable overall top1 accuracy.

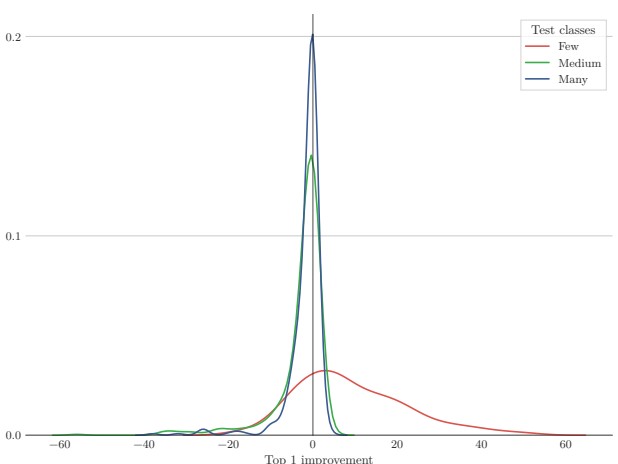

Figure 4: Improvements in average Alpha cRT (best few) top 1 accuracy for each class in the test set, grouped as "few", "medium", or "many" classes. A KDE estimator was fit for each group shown.

We now look at per-class accuracy; Figure 3a shows our class-wise top 1 improvement compared to state-of-the-art, where each bar indicates a separate class in "few". As shown, we observe an overall large improvement for each class. Simultaneously, in Figure 4 we show a KDE estimator that was fitted on the improvements in "few", "many", and "medium". We note that the average class decline in "many" and "medium" classes are constrained to low numbers while a large number of strong classifiers retain original top 1 class performance.

Furthermore, we examine the relationship between the improved "few" classes and their nearest neighbor classifiers. In Figure 3b we show each "few" class improvement and the corresponding euclidean distance of the class' closest neighbor. We observe that the closer the neighbor, the better the average class improvement, and vice versa. We believe this demonstrates the power of composing new classifiers with neighbors that are highly correlated with the target class.

Finally, we present results on Places-LT in Table 2. As only Resnet-152 models were available for cRT and LWS trained on Places-LT, we present our results on these models. We outperform cRT and LWS by a respective 2.1 and 1.4 in the "few" split.

**Exploration of** $\gamma$ We explore the hyperparameter $\gamma$ and its effects on Alpha Net. As Alpha Net learns the coefficients for linearly combining classifiers, the limit of $\gamma$ forcefully restricts the contribution of the original weak classifier to the final composition. We set the range of possible $\gamma$ from $[0.1, \ldots, 0.9]$, where the higher the $\gamma$, the more the original weak classifier is weighted. We show our results on all $\gamma$ values for ResNeXt-50 cRT using ImageNet-LT in Figure 5. We note that while $\gamma$ is a hyperparameter that affects the results, cross-validation *consistently* selects $\gamma = 0.7$ for both ImageNet-LT and Places-LT, with the exception of $\gamma = 0.6$ for Alpha-LWS on Places. More importantly, we consider $\gamma$ to be our 'control' for performance trade-off. As observed in Figure 5 if one wants to preserve and improve upon overall results while still improving on the weak classifiers, a higher $\gamma$ is appropriate. If a higher improvement on weak classifiers is prioritized, a lower $\gamma$ is able to achieve that while obtaining comparable overall performance. Our results reaffirm our assertion

| Methods | Few | | Medium | | Many | | All | |
|---|---|---|---|---|---|---|---|---|
| | top1 | top5 | top1 | top5 | top1 | top5 | top1 | top5 |
| Joint† | 8.2 | - | 27.3 | - | 45.7 | - | 30.2 | - |
| NCM† | 27.3 | - | 37.1 | - | 40.4 | - | 36.4 | - |
| t-normalized† | 31.8 | - | 40.7 | - | 37.8 | - | 37.9 | - |
| cRT† | 24.9 | - | 37.6 | - | 42.0 | - | 36.7 | - |
| LWS† | 28.6 | - | 39.1 | - | 40.6 | - | 37.6 | - |
| cRT∓ | 24.3 | 55.0 | 36.7 | 69.5 | 41.3 | 72.1 | 35.9 | 67.6 |
| LWS∓ | 27.7 | 58.5 | 38.3 | 70.1 | 39.1 | 72.0 | 36.5 | 68.5 |
| Alpha - cRT | **26.4** | **59.6** | 35.1 | 67.7 | 40.6 | 71.5 | 35.4 | 67.5 |
| Alpha - LWS | **29.1** | **60.7** | 37.0 | 68.8 | 38.6 | 71.3 | 36.0 | 68.1 |

Table 2: Places-LT top1 and top5 accuracy for Alpha Net. Methods with † indicate that the results were directly taken from Kang et al. (2020). Methods with ∓ indicate that the results were reproduced with the models from Kang et al. (2020) but did not achieve the reported results.

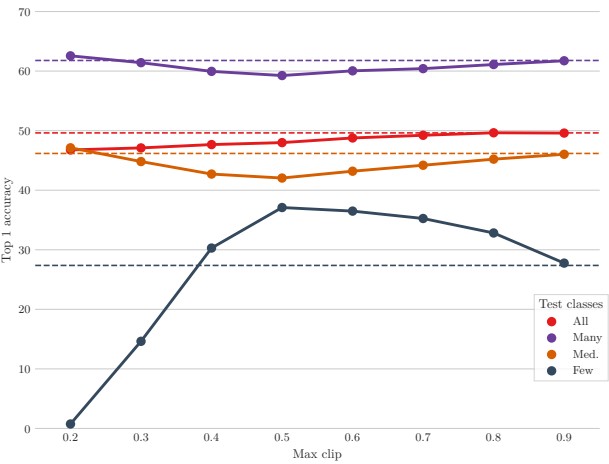

Figure 5: Alpha cRT top 1 accuracies on Imagenet-LT with varying max clip, $\gamma$. Dotted lines show baseline cRT accuracies.

that while a combination of relevant strong classifiers can improve a weak classifier, a composition without including the original weak classifier leads to poor results. Indeed, qualitatively, what Alpha Net is doing is adjusting the original position of the weak classifier towards the appropriate strong classifiers. However, without the weak classifier as a starting point, any combination of the strong classifiers becomes a possible - and less effective - solution.

## 5 CONCLUSION

We present a novel model, Alpha Net, which showcases the ability to compose better performing classifiers in model space. Specifically, Alpha Net is able to linearly combine strong classifiers to adaptively adjust a weak classifier. Our method demonstrates the power of working within classifier space and affirms that - at least for some vision problems - classifier space is a more efficient space to work in as compared to model feature space. With Alpha Net, we are able to establish a new benchmark for state-of-the-art performance on the "few" classes within standard long-tailed recognition datasets. Finally, we believe that Alpha Net's capability to run on any classification model suggests its future potential to improve upon general classification tasks beyond long-tailed recognition.

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

## A    ALPHA NET WEIGHT SHARING

Alpha Net consists of a sub-module for every new class for which one wishes to compose a new classifier. With every additional new class, Alpha Net's memory and computation time linearly increases. We explore the possibility of weight sharing in Alpha Net to address this issue. Note that each alpha block consists of 2 fc layers; as such, we report results on Alpha Net where both fc layer weights are shared across all alpha blocks. In Figure 6, we observe a slight drop in best accuracy as compared to Alpha Net with multiple alpha blocks. This showcases Alpha Net's ability to compose new classifiers without large computation and memory overhead. However, this increase in efficiency comes with a cost: by weight sharing, Alpha Net loses the flexibility to adjust each sub-module so as to maximally benefit that class. As an example, some classes may benefit from dynamically changing the inputs or the number of nearest neighbor classifiers that are combined. Thus, the advantage of multiple alpha blocks is class-wise flexibility and versatility in training dynamics.

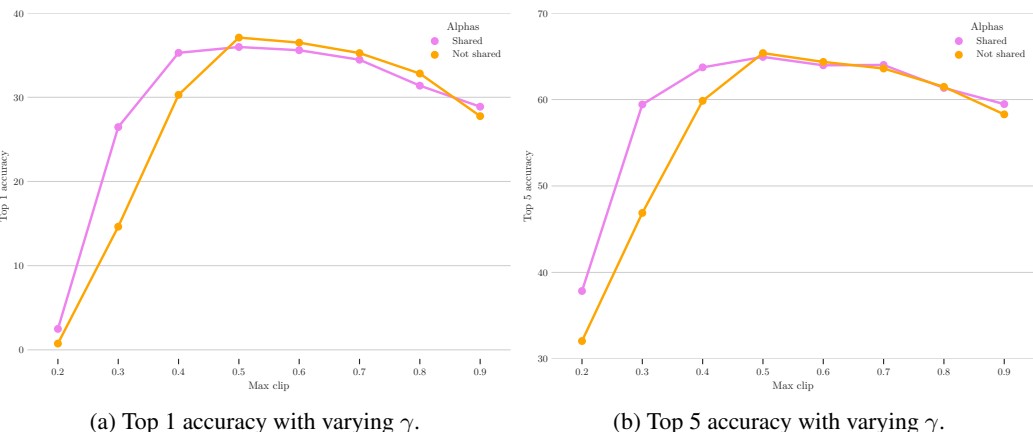

(a) Top 1 accuracy with varying $\gamma$.          (b) Top 5 accuracy with varying $\gamma$.

Figure 6: Evaluation of weight sharing for Alpha Net with varying $\gamma$. Alpha cRT accuracies in both figures come from the ImageNet-LT "few" test set. "Not shared" denotes Alpha Net trained with separate alpha blocks' weights as used in the main paper. "Shared" denotes Alpha Net with weights shared across all alpha blocks. See the text for discussion.

## B    VISUALIZATION OF LEARNED ALPHA VALUES

To better understand the classifier composition, we show the learned alpha coefficient values for our best Alpha cRT model trained on ImageNet-LT. As seen in Table 3, each set of alphas for each class is different. While we observe a consistent high alpha learned for the original classifier, the distribution of the remaining alphas varies across different classes. Of note, we see that the immediate nearest neighbor (denoted as "NN1") does not always have the highest alpha. In fact, the highest alpha changes within the top $k$ depending on the class.

| NN1 | NN2 | NN3 | NN4 | NN5 | Original |
|---|---|---|---|---|---|
| 0.2941 | 0.0600 | 0.0600 | 0.0600 | 0.0600 | 0.7000 |
| 0.1672 | 0.1326 | 0.0600 | 0.0600 | 0.0600 | 0.6996 |
| 0.0600 | 0.2995 | 0.0600 | 0.0600 | 0.0600 | 0.7000 |
| 0.0600 | 0.0600 | 0.1534 | 0.1458 | 0.0600 | 0.7000 |
| 0.1081 | 0.0600 | 0.2358 | 0.0600 | 0.0600 | 0.6560 |
| 0.0600 | 0.0600 | 0.0600 | 0.1760 | 0.1238 | 0.7000 |
| 0.0600 | 0.0600 | 0.0600 | 0.2997 | 0.0600 | 0.7000 |

Table 3: Sets of learned alphas from our Alpha cRT model for 10 classes on ImageNet-LT. Each row represents a set of alphas for a single class, where nearest neighbor 1 ("NN1") is the top 1 NN's classifier and "original" is the original classifier of that class, respectively. Alphas are taken from the model reported in Table 1 trained with $\gamma = 0.7$ and the same hyperparameters listed in Section 4 of the main paper.

## C  TOP $K$ NEAREST NEIGHBOR STUDY

We explore the effect of the number of top $K$ nearest neighbor classifiers used in Alpha Net. We varied our top $K$ from (2,5,10) and show our results on ImageNet-LT with Alpha cRT in Figure 7. Interestingly, there is no significant accuracy difference when we increase our $k$ from 5 to 10. However, we note a large decline in performance when we limit $k$ to only 2. This decline demonstrates the importance of combining a sufficient number of classifiers: using an insufficient number of classifiers limits the capability to adjust our classifier.

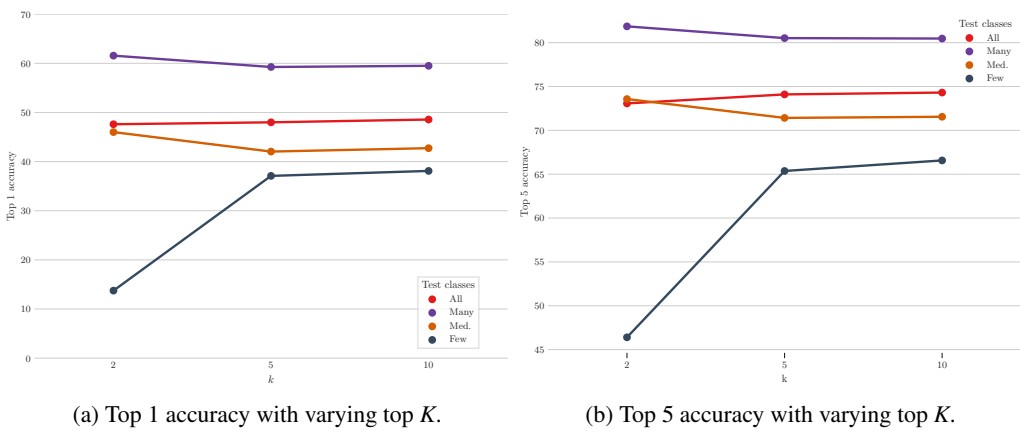

(a) Top 1 accuracy with varying top $K$.  (b) Top 5 accuracy with varying top $K$.

Figure 7: Effects of top $K$. Alpha cRT accuracies in both figures come from the ImageNet-LT test set, grouped as "few", "medium", or "many" classes.

## D  WHERE TO SAMPLE TOP $K$ CLASSIFIERS?

In all experiments shown in Section 4 of the main paper, we sampled our top $K$ nearest neighbors from the "base" classes, which we define as the combination of "many" and "medium" splits. Here, in addition to "base", we explore the effect of sampling top $K$ from alternative splits. Specifically, we show results of using splits "many" and "all", where "all" is the combination of all splits – "many", "medium", and "few". In Figure 8, we observe noticeable accuracy drop when using only "many" split, as compared to using "base" split. We attribute this result to the fact that "base" split has twice the number of classes to select from, thus enabling better nearest neighbor possibilities. Finally, we do not see a significant difference between "all" split versus "base" split. "All" split contains the additional classes from the "few" classes, which have weak classifiers. Thus, potential freedom to choose weak classifiers does not provide additional benefits to achieve our main goal of composing a new classifier from strong classifiers.

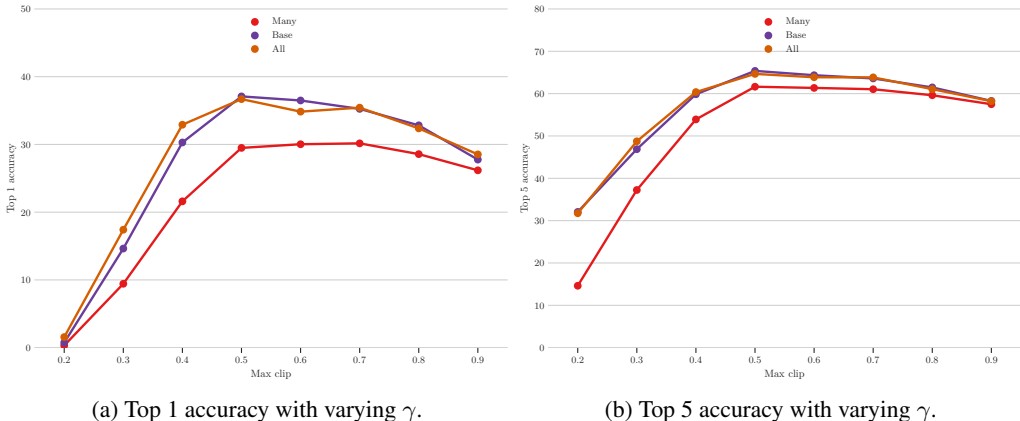

(a) Top 1 accuracy with varying $\gamma$.  (b) Top 5 accuracy with varying $\gamma$.

Figure 8: Evaluation of class splits from which we sample top $K$ classifiers over all gammas. Alpha cRT accuracies in both figures come from the ImageNet-LT "few" test set. The top $K$ classifiers are sampled from "many", "base", or "all" classes.

## E NEGATIVE TRAINING SAMPLES

All experiments in Section 4 of the main paper used both equal amount of positive and negative "few" training samples for Alpha Net. Here we remove all negative training samples to explore its effects on accuracy. We showcase our results in Table 4. With only positive training samples, we observe a higher "few" top 1 accuracy compared to training with equal amount of positive and negative samples. However, we simultaneously see a larger decrease of 3.2% in overall top 1 accuracy. This result illustrates that training with only "few" positive samples will cause Alpha Net to compose a better classifier for "few" classes at a higher expense of the overall top 1 performance. In order to ensure good overall top 1 performance, we suggest training with negative samples as well.

| Negative Samples | Few | | Medium | | Many | | All | |
|---|---|---|---|---|---|---|---|---|
| | top1 | top5 | top1 | top5 | top1 | top5 | top1 | top5 |
| No | **40.1** | 66.9 | 39.6 | 70.9 | 57.1 | 80.0 | **46.4** | 73.8 |
| Yes | **37.1** | 65.4 | 42.0 | 71.4 | 59.3 | 80.5 | **48.0** | 74.1 |

Table 4: Impact of negative training samples for Alpha Net on ImageNet-LT. Models displayed are trained with $\gamma = 0.5$ and the same hyperparameters listed in Section 4 of the main paper.

## F ADDITIONAL EXPLORATIONS OF $\gamma$

In Figure 6 of the main paper, we showed the top 1 accuracy for ImageNet-LT with varying $\gamma$ and top $K$. Here, we first display the accompanying top 5 accuracy results in Figure 9. In addition to varying $\gamma$ for Alpha cRT, we also show the best $\gamma$s' effects on Alpha LWS in Figure 10. We observe that $\gamma$'s effects on Alpha cRT and Alpha LWS are similar. We observe that $\gamma$'s effects on Alpha cRT and Alpha LWS are similar. The few accuracy consistently peaks at $\gamma = 0.6$, and the higher or lower $\gamma$, the higher or lower the accuracy. Lastly, in Figure 11 and Figure 12 we show the $\gamma$s' effects for Alpha cRT and LWS respectively on Places-LT. While the improvement in "few" is not as high as that in ImageNet-LT, we again observe that $\gamma$s' effects are consistent.

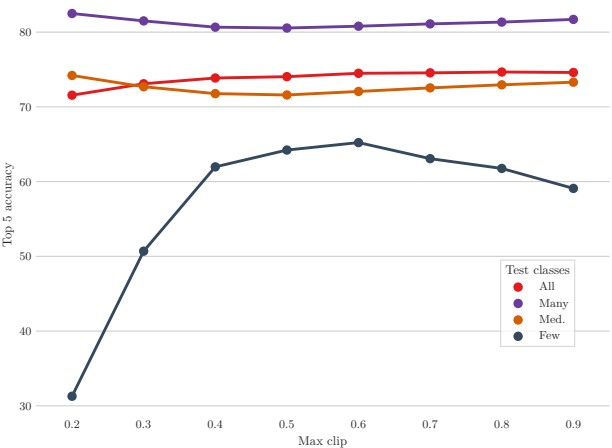

Figure 9: Top 5 accuracy with varying max clip $\gamma$. Alpha cRT accuracies come from the ImageNet-LT test set, grouped as "few", "medium", "many", or "all" classes.

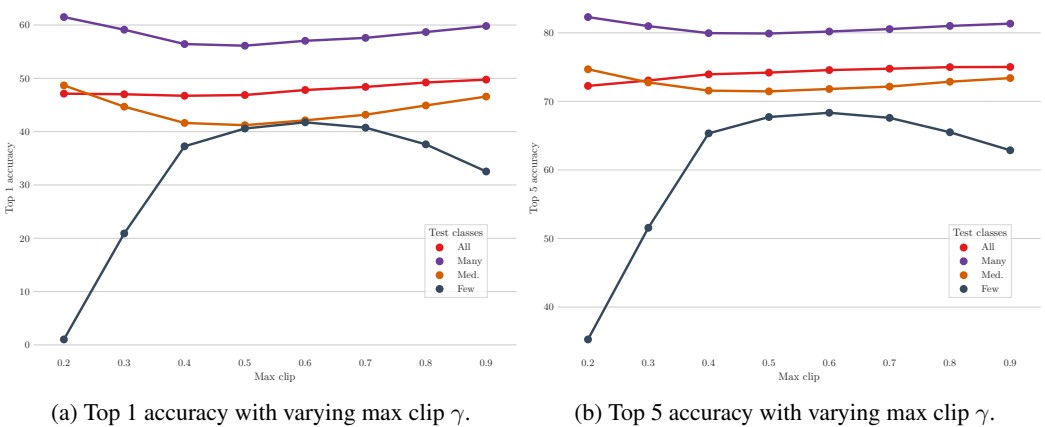

(a) Top 1 accuracy with varying max clip $\gamma$.

(b) Top 5 accuracy with varying max clip $\gamma$.

Figure 10: Effects of max clip $\gamma$. Alpha LWS accuracies in both figures come from the ImageNet-LT test set, grouped as "few", "medium", "many", or "all" classes.

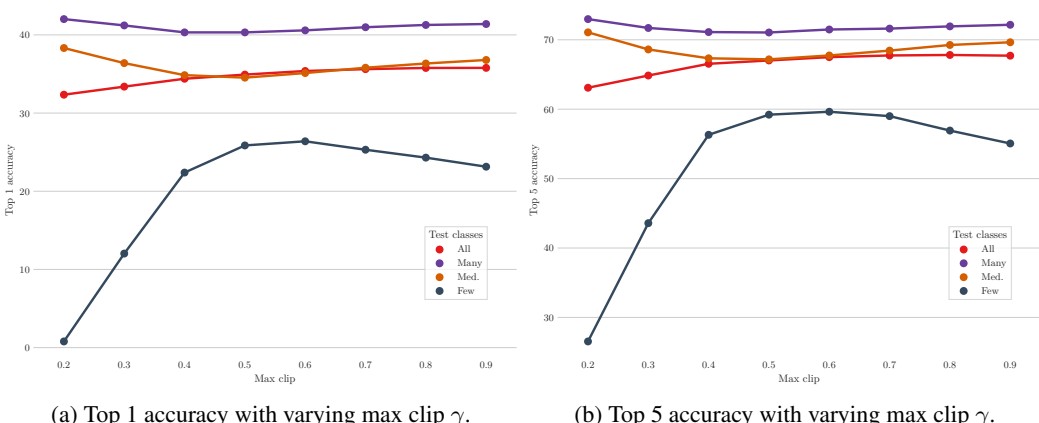

(a) Top 1 accuracy with varying max clip $\gamma$.

(b) Top 5 accuracy with varying max clip $\gamma$.

Figure 11: Effects of max clip $\gamma$. Alpha cRT accuracies in both figures come from the Places-LT test set, grouped as "few", "medium", "many", or "all" classes.

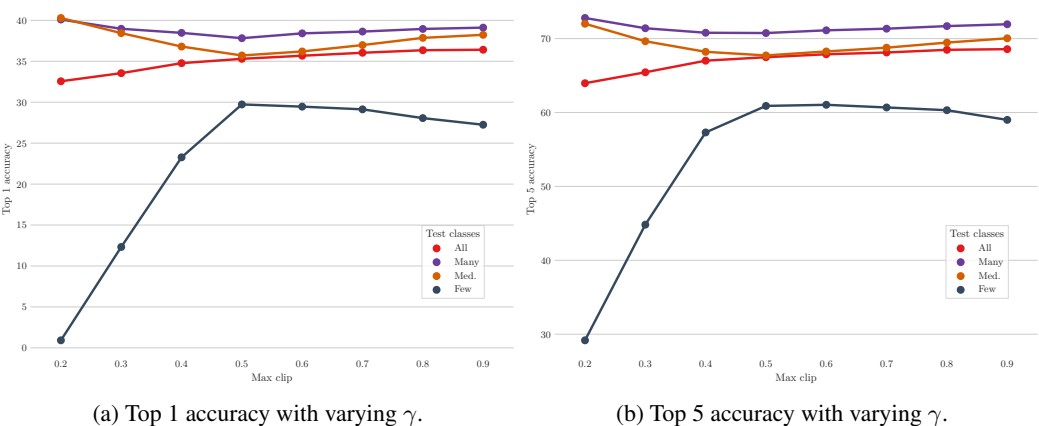

(a) Top 1 accuracy with varying $\gamma$.        (b) Top 5 accuracy with varying $\gamma$.

Figure 12: Effects of max clip $\gamma$. Alpha LWS accuracies in both figures come from the Places-LT test set, grouped as "few", "medium", "many", or "all" classes.

