# OpenReview forum: "Alpha Net: Adaptation with Composition in Classifier Space"
_ICLR.cc/2021/Conference — Reject_

### Official Review · AnonReviewer2 · 2020-10-27
**The proposed method appears to have technical flaws. I don’t think that the work is ready to be accepted for publication.**

**Rating:** 3
**Confidence:** 5

**Review:**

This paper addresses the well-known long-tail classification problem. The argument made here is that most of the existing methods attempt to transfer knowledge in the feature space, which is true. Based on this motivation, the paper proposes a method to do the knowledge transfer in the model space instead. The idea is to apply K-NN method in the model space to pick up a group of strong classifiers trained on the head classes with sufficient training samples available that are closest to a weak classifier in the model space and then a linear combination of the group of strong classifiers and the weak classifier to form a stronger classifier to the tail classes where only few samples are available for training; the linear combination weights are learned from a simple neural network, called Alpha Net. Two datasets artificially truncated from ImageNet and Places, respectively, that were also used in the peer work in the literature, were used to report the evaluations.

The paper reads very well, except for a few grammatical errors. The presentation is clear and easy to follow.

My major comments follow. The long-tail learning problem is not new, and the idea of knowledge transfer in the model space is not new either (e.g., King et al 2019 referenced in the paper and in fact that reference was updated in 2020 with better results beating what this paper reported). Consequently, the novelty of this work is rather limited.

Further, I have a strong reservation in considering the proposed method as a technically sound approach. Conceptually, the idea of combining a group of closest strong classifiers with a weak classifier to form a stronger classifier in the model space is based on the proximity presumption. Regretfully, unlike in the feature space where the proximity presumption is valid in general (unless the feature points are located close to the class boundaries), I am not convinced that the same proximity presumption is valid in the model space, as it is easy to give many counter examples.

Regarding to the experiments, I would like to mention that the authors of the closest competitor, Kang et al 2019 referenced in the paper, have updated their work in arXiv this year with results beating what was reported in the paper. Also they used more datasets to evaluate their method than the two datasets used in the paper. So it is difficult to argue that the proposed method represents the state-of-the-art.

Overall, I am not convinced that the proposed method is technically sound and advances the state-of-the-art literature.

---

I appreciate the authors' effort in responding to my comments. But the arguments in their response appear in conflict. Overall, I am still not convinced by their arguments. So I stay with my original review.

---

> ### Author Response · Authors · 2020-11-25
> **Response to Reviewer 2**
>
> We thank the reviewer for their comments.
>
> 1) We would like to note that contrary to the reviewer's statement, we do use the updated 2020 results from Kang et al. In their 2019 arXiv version, Kang et al did not provide any LWS models. In our paper we compare against LWS models found in the 2020 version of their paper as well as all other models and performances found in the 2020 version.
>
> 2) We acknowledged in the paper that long-tail learning and knowledge transfer in model spaces are not new. The reviewer states that Kang et al does knowledge transfer in model space. To the best of our knowledge, this is not the case. Finally, we are unaware of any other paper that has applied classifier combination to long-tailed learning.
>
> 3) Regarding the reviewer’s statement that “proximity presumption is not valid in model space”, we note that several related works do rely on this presumption for their methodology, especially for zero-shot learning [1,2]. One of them, as cited in the main paper, by Aytar & Zisserman [2], utilizes regularized minimization to combine SVMs. Finally, we note that there is no easy way to combine features as we have done for classifiers. Since features (from the penultimate layer of networks) are generated from the same set of weights, they cannot be adapted for new classes without also modifying the representation for previous classes. However, since the classifier, i.e. the last layer, has a separate weight vector for each class, we can add new classes (by specifying their classifier weights) independently.
>
> 4) We presented results on two of the three datasets used by Kang et al., Imagenet-LT, and Places-LT. We did not initially include the 3rd dataset iNaturalist in our results because while iNaturalist is a natural long-tailed dataset, the tail performance on iNaturalist does not follow expected behavior for long-tailed dataset. Specifically, the cRT model is able to achieve an accuracy of 61.0 for many, 63.5 medium, 63.3 for few, and 63.1 for all. These numbers are very different compared to that of a normal long-tailed dataset. In ImageNet, the same model achieves 61.8 for many, 46.2 for medium, 27.4 for few, and 49.6 for all classes. We note that in ImageNet-LT there is a gap of a large 34.4 points between many and few, whereas in iNaturalist the few classes in fact outperform many classes by 2.2 points. This unexpected behavior and high performance on tail classes makes iNaturalist not a suitable dataset for our methodology, which intends to improve the low accuracy tail classes. Due to computation and time limits, we are not able to complete iNaturalist experiments at this time. However, we will include iNaturalist in the next revision.
>
> 5) In regards to advancing state-of-the-art literature, as stated in the main paper we only claim to significantly advance state-of-the-art on the tail classes of a long-tailed dataset, as they are the most challenging classes of the dataset.
>
> References
> 1. Mensink, Thomas, Efstratios Gavves, and Cees GM Snoek. "Costa: Co-occurrence statistics for zero-shot classification." Proceedings of the IEEE conference on computer vision and pattern recognition. 2014.
> 2. Aytar, Yusuf, and Andrew Zisserman. "Enhancing Exemplar SVMs using Part Level Transfer Regularization." BMVC. 2012.

---

### Official Review · AnonReviewer3 · 2020-10-28
**The paper presents a method for robustly handling long-tailed learning and demonstrated the impact on image classification.**

**Rating:** 8
**Confidence:** 4

**Review:**

Significance:
This article is a useful contribution to transfer learning for tasks where there is not enough data available, showing a modest improvement over the other methods that employ transfer learning in the classifier space.

Novelty:
The main contribution of this paper is the improvement of weak classifiers when there is not enough data for a class by combining the weak classifiers with the most relevant strong classifiers. This method finds k closest strong classifiers to the weak classifier and then combines the weak classifier with existing classifiers without creating new classifiers or networks from scratch.

Potential Impact:
The approach presented in this paper is well-evaluated in computer vision, but potentially useful in many other settings.

Technical Quality:
The technical content of the paper appears to be correct.

Presentation/Clarity :
The paper is generally well-written and structured clearly. While this method is a clear winner on Few classes, it is not performing as well in Medium classes, as shown in Table 1. An explanation about this issue could strengthen the paper.

Reproducibility:
The paper describes all the algorithms in full detail and provides enough information for an expert reader to reproduce its results. I would suggest the authors release their code on GitHub or other sites to help other researchers reproduce their results.

---

> ### Author Response · Authors · 2020-11-25
> **Response to Reviewer 3**
>
> We thank the reviewer for their encouraging review.
> 1) We will release our code on GitHub as suggested.
> 2) Regarding the medium classes performance, consider Figure 5 from the main paper. The figure illustrates that at a gamma regularizer around 0.8, we can achieve similar medium performance. While the few performance improvement is still substantial, it is not its maximal performance. To ensure as high of a performance as possible for few classes, our regularizer slightly lowers performance on medium classes.

---

### Official Review · AnonReviewer1 · 2020-11-01
**This paper focuses on how to transfer knowledge between classes. The authors proposed to transfer classifiers instead of features. The proposed to linearly combine the classifiers from rich classes to construct more robust classifiers or rare classes. The combination weights are predicted from a learned neural network for each rare class. The experimental results on two benchmark datasets outperform some existing methods. The paper, however, needs some more comparison, analysis, and discussion.**

**Rating:** 4
**Confidence:** 5

**Review:**

# Summary
This paper focuses on how to transfer knowledge between classes. The authors proposed to transfer classifiers instead of features. The proposed to linearly combine the classifiers from rich classes to construct more robust classifiers or rare classes. The combination weights are predicted from a learned neural network for each rare class. The experimental results on two benchmark datasets outperform some existing methods.

# Strengths
- While the idea of constructing classifiers by a linear combination of other classifiers has been proposed for several different problems (e.g., zero-shot learning), its application to long-tailed classification seems to be novel.

- The idea is clean and clear. The approach part is well-written.

- The proposed method improves the performance of tail classes.

# Weaknesses
1. There is no comparison to existing methods (proposed in other problems) that use linearly combine classifiers. Note that, those methods can be applied to long-tailed classification with minimal modifications. For example, for ZSL, due to the lack of visual information of unseen classes, the combination weights can only be estimated from the semantic descriptions. Here, with the W_j for each class, one baseline is thus to replace the semantic descriptions by W_j or even visual features for estimating the combination weights. The authors should compare to those methods.

2. While the approach is clearly written mostly, the need for independent alpha-net modules for each tail class is unclear. Note that, an alpha-net is a two-layer network with lots of parameters, and for tail classes, there are only a few labeled data instances. Learning for each class an alpha-net may be vulnerable to over-fitting.

- There is no equation of the training loss for the alpha-nets. It will be great to provide it. If I understand correctly, the loss is still a softmax loss across all the classes (head and tail), but only the alpha-net parameters (for the tail classes) are being learned.

- The alpha-nets seem to be learned from the data that have been used to train the classifiers in the first stage. As neural networks can usually achieve very low training error, alpha-nets with a one-hot output (i.e., only use V^j_0) may already lead to very high training accuracy. I’m not sure if alpha-nets learn anything meaningful.

- An ablation study on the algorithm design, for example, using a shared alpha-net for all the classes, should be included.

3. The related work and experimental comparison are insufficient. Only “one” paper has been compared in Table 1 and Table 2. There have been many papers published in CVPR 2020 and ECCV 2020. The authors, however, cited NO papers published in 2020.

4. Can the authors provide more discussions on why the performance at medium, many, and all classes drop in comparison to the baselines? For now, it seems that the performance improvement comes simply by trading the prediction/adjusting the classifier strengths among classes: for example, increasing the classifier “norms” of tail classes.

# Minor
- The figures and captions can be improved. Specifically, Figures 1 and 2 are not self-contended: it is hard to understand the figures without looking at the main text.

# Justifications
While the proposed idea seems novel for long-tailed classification, the paper lacks comparisons to existing methods and comparisons to similar algorithms proposed in other problems (with minimal changes). There is no ablation study on why we need an alpha-net module for each class. There is no overall performance gain, making it hard to tell if alpha-nets really improve classifiers or simply trading predictions/adjusting the classifier strengths among classes. I thus give a score of 3.

----------------------------- Post rebuttal -----------------------------

I read the author's rebuttal and I greatly appreciate their efforts. The authors have done many more experiments and I would like the authors to incorporate them into their manuscript and modify their manuscript, even methods, accordingly. I think these new materials can greatly strengthen the paper.

1) It seems that ZSL with the original classifier involved is quite strong (this could not happen in ZSL as there is no original classifier for the unseen classes). I would suggest that the authors further investigate this for a detailed comparison. These methods may even simplify the authors' methods, and a connection to ZSL can strengthen the paper. For instance, Changpinyo et al., 2016) showed that their method can outperform [1] and it will be interesting to have some further comparison.

2) It's nice that the authors compare the shared and non-shared alpha net. I still have doubt that why non-shared alpha net won't over-fit given that there are only a few labeled data instances. A shared alpha net might be more suitable for robustness.

3) There is one difference to Kang's method. Kang's first stage stopped earlier so tailed classifiers have not covered. Did the author do the same thing?

4) One method that can simply trade-off the accuracy is Kang's method. I think you can tune their hyperparameter to get a higher tail accuracy. Now the question will be, what will their head accuracy be? Without having a more ground comparison among methods, my question still remains unsolved.

5) Besides ImageNet-LT and PlaceNet-LT, there are several CVPR/ECCV papers that outperform Kang's paper on CIFAR, iNaturalist but do not report on these two datasets.

I have increased the score to 4, but I think the paper needs significant work to incorporate my comments as well as other reviewers' comments to be ready for being published.

---

> ### Author Response · Authors · 2020-11-25
> **Response to Reviewer 1**
>
> We thank the reviewer for their comments.
> 1) As suggested we compared to a zero-shot learning (ZSL) classifier combination where we replace semantic descriptions with visual features. In a baseline ZSL paper [1], they estimate a new weight for an unseen class by linearly combining existing classifiers. The coefficients used for combination is obtained through a similarity score, which relies on auxiliary data. Since we do not have access to auxiliary data, we use visual features as also suggested by the reviewer. By using the normalized cosine similarity scores between averaged class features, we are able to create a set of coefficients to combine the top 5 nearest neighbor (NN) classifiers. Furthermore, for the best comparison to Alpha Net we similarly combine the original classifier (W) with the top 5 NN classifiers. Critically, we note that without including the W, the baseline performance is worse. Thus, including W is also necessary for linear combinations to succeed. For Alpha Net we observe the best results with the W coefficient set at 0.7, which we also use as the coefficient in this case. The remaining top 5 classifiers are normalized such that they all sum to (1 - 0.7), 0.3. This baseline achieves top1 accuracies on ImageNet-LT: many: 62.45, medium: 46.99, few: 15.29, all: 48.63. Our Alpha Net has top1 accuracies of many: 60.5, medium: 44.3, few: 34.5, all: 49.2. We significantly outperform the baseline few performance by 19.21 points, and our overall is higher by 0.57 points. Conclusively, Alpha Net is able to achieve better performance than the baseline ZSL classifier combination method.
> 2) As suggested, we have implemented weight sharing on all alpha blocks. We have added a plot showing the performance of a shared Alpha Net compared to a non-shared Alpha Net. As illustrated in Fig. 6, we see that a non-shared Alpha Net has higher performance than as the shared Alpha Net. Critically, a shared Alpha Net also illustrates that gamma stabilizes around 0.7 with a similar pattern as that of the non-shared Alpha Net. Thus, if computation is a constraint, a shared Alpha Net is still capable of somewhat similar results. Overall, the stability of shared Alpha Net is encouraging and exemplifies the efficiency of our method.
> 3) Regarding the usage of the same data to train for Alpha Net, we note that Kang et al. also use the same training data used for pre-training when tuning their classifier. In their case, they use a resampling strategy to focus more on improving tail classifiers. Similarly, Alpha Net focuses on tail samples and we too improve on the tail.
> 4) Fig 5 in the main paper shows our regularizer gamma ablation. The figure shows that a 0.8 gamma does affirm Alpha Net’s ability to produce similar many and medium performance as that of baseline while gaining a significant improvement in few. Note that the few classes are the most challenging classes, as exemplified by the low accuracy for tail classes in all methods. Additionally, we know of no other approach which can achieve as high performance on the tail classes, while sacrificing very little in the head classes. The lack of such an approach indicates that improving tail classifiers, even by trading predictions, is an extremely difficult problem. We see no obvious way to improve the tail classifiers in such a high dimensional space.
> 5) We cited the ArXiv version of Kang et al., but we note that it is a ICLR 2020 paper and we compare to its 2020 results. Regarding CVPR and ECCV 2020 papers, we note that only 1 ECCV paper [2] has better performance than Kang et al. However, their method is extremely complex and relies on a class tree hierarchy with two regularizers. Thus, their improvement is mainly over iNaturalist where there is an inherent hierarchy to exploit. We note that in Kang et al. the few performance for iNaturalist is as high as the many classes. Their few performance for ImageNet-LT is not included. Further, we note that 1 ECCV 2020 [3] and 1 CVPR 2020 [4] paper have worse performance than Kang et al. No papers in CVPR 2020 and ECCV 2020 have methods that are as simple as Kang et al. and have as good of a performance.
> 6) The reviewer is correct in that only the Alpha Net parameters for tail classes are being learned and the loss is a cross entropy loss for all classes. We will add the loss and improve captions in the next revision.
>
> References
> 1. Mensink, Thomas, Efstratios Gavves, and Cees GM Snoek. "Costa: Co-occurrence statistics for zero-shot classification." CVPR. 2014.
> 2. Wu, Tz-Ying, et al. "Solving Long-tailed Recognition with Deep Realistic Taxonomic Classifier." ECCV. Springer, Cham, 2020.
> 3. Xiang, Liuyu, Guiguang Ding, and Jungong Han. "Learning from multiple experts: Self-paced knowledge distillation for long-tailed classification." ECCV. Springer, Cham, 2020.
> 4. Jamal, Muhammad Abdullah, et al. "Rethinking Class-Balanced Methods for Long-Tailed Visual Recognition from a Domain Adaptation Perspective." CVPR. 2020.

---

### Official Review · AnonReviewer5 · 2020-11-06
**Interesting idea that boosts tail classifiers but hurts overall performance**

**Rating:** 4
**Confidence:** 4

**Review:**

============================================

Final recommendation after rebuttal

The authors gave a good rebuttal, and the current version of Fig 5 and the new Fig 6 are making the paper stronger in my opinion.  However, I will stick to my previous rating as:
a) the main weakness, the tradeoff in performance for few vs overall makes the contributions weak, especially since the cRT baseline is as simple as finetuning only the classifers with balanceed sampling. I would expect gains over that

b) Fig 6 further shows the marginal gains over a weight-sharing baseline and makes the basic approach questionable

c) there are no experiments on real-life long tailed datasets - a note here about iNaturalist: the argument the authors make about iNat makes some sense to me, and I want to thank them for replying. But this is exactly why we should test on real long-tail datasets, ie they dont behave as the artificially created ones.

============================================



The paper presents an interesting idea, transferring of knowledge between head and tail classes at the classifier level, ie create stronger classifiers for the tail classes by linear combinations of  a tail classes' nearest neighbour classifiers with the current "weak" one.

In general, although interesting conceptually, the approach doesn't seem to work better than the baselines overall, and the paper doesnt offer any further interesting analysis or insights for long-tailed recognition that would make the performance part be negligible.

Strengths:

*Long-tail recognition and learning from imbalanced data is an interesting, realistic and important problem
* The authors propose an approach that helps learn better few shot classifiers increasing the performance on the tail classes, trading it off with slightly reducing performance overall

Weaknesses:
* The authors compare to strong baselines, and do indeed increase performance for tail classes, but in the end they harm overall performance. To get the "10%" margin mentioned in the abstract, they also reduce overall performance by 1.5% and head-class performance by around 3%.
* It seems there is a tradeoff here, where to learn better tail classifiers you hurt head class performance. Would another hyper-parameter setup give the same med/many performance (not harm) and increase few shot performance? Maybe baseline performance (as horizontal lines for few/med/many/all) could be included in the hyperparameter exploration plots.
* The authors do not present results on any real long-tail dataset, eg iNaturalist, but only on two smaller and artificially created LT datasets (which are the standard, but also many times accompanied with results on a real long-tailed dataset like iNat or faces)
* The clipping hyperparameter $\gamma$ seems to control performance a lot, more than the number of classifiers combined (K). Controlling this with a clamping parameter seems heuristic without further discussion.  Although the parameters are ablated,  and seem relatively stable, it is not discussed why clamping is so important and why  "we consider γ to be our ‘control’ for performance trade-off".
* Given that the input to the added 2-layer network Alpha-net is the ordered list of the K closest classifiers and the weak classifier, it is unclear why the authors choose to have one $A_k$ model per class.  What would performance be if there was a single network for all classes? This is a missing baseline that is kind of needed to justify the added computational complexity.

Notes:
* From the "three advantages" enumerated in Sec1, I dont understand how the second is an advantage over other approaches; to me it is more of the way to make this approach work. Same with the third advantage; the coefficients are learned for this method - I dont understand how it is an advantage that they are learned more adaptively when related works dont have those coefficients in the first place.
* It is unclear what Figure 4 offers and it is hard to comprehend. Some more analysis (or a citation) on what the kernel density estimator is is needed
* Do all $\alpha_i$'s for a weak classifier sum to 1 after clamping?

---

> ### Author Response · Authors · 2020-11-25
> **Response to Reviewer 5**
>
> We thank the reviewer for their comments.
>
> 1) We did not initially include iNaturalist in our results because while iNaturalist is a natural long-tailed dataset, the tail performance on iNaturalist does not follow expected behavior for long-tailed dataset. Specifically, the cRT model is able to achieve an accuracy of 61.0 for many, 63.5 medium, 63.3 for few, and 63.1 for all. These numbers are very different compared to that of a normal long-tailed dataset. In ImageNet, the same model achieves 61.8 for many, 46.2 for medium, 27.4 for few, and 49.6 for all classes. We note that in ImageNet-LT there is a gap of a large 34.4 points between many and few, whereas in iNaturalist the few classes in fact outperform many classes by 2.2 points. This unexpected behavior and high performance on tail classes makes iNaturalist not a suitable dataset for our methodology, which intends to improve the low accuracy tail classes. Due to computation and time limits, we are not able to complete iNaturalist experiments at this time. However, we will include iNaturalist in the next revision.
>
> 2) Gamma is a way of regularizing the problem. As a regularizer, gamma effectively narrows the solution space of the final classifier, such that it is close to the original solution of the classifier vector. By controlling gamma, we control how far the solution is allowed to move from the original classifier space. For example, in our gamma ablation if we increase gamma such that gamma approaches 1, we effectively have the same original classifier. On the other hand, as gamma approaches 0, we have worse performance because we are changing the classifier entirely without any regularizer to enforce a sensible solution space. We select gamma using performance on the validation set, as is common practice for regularizers [1].
>
> 3) As suggested we have updated Fig 5 in the main paper to include baseline numbers. The figure shows that a gamma of 0.8 does indeed affirm Alpha Net’s ability to produce many and medium performance similar to that of baseline while gaining significant improvement in few. Note that the few classes are the most challenging classes in a long-tailed dataset to improve. We know of no other approach which can achieve as high performance on the tail classifiers, while sacrificing very little in the head classes. We see no obvious way to improve the tail classifiers in such a high dimensional space.
>
> 4) Thank you for suggesting weight sharing. As suggested, we have implemented weight sharing on all alpha blocks. We have added a plot showing the performance of a shared Alpha Net compared to a non-shared Alpha Net. As illustrated in Fig. 6, we see that a non-shared Alpha Net has higher performance than the shared Alpha Net. Critically, a shared Alpha Net also illustrates that gamma around 0.7 is optimal, with a similar pattern as that of the non-shared Alpha Net. The stability of shared Alpha Net is encouraging and exemplifies the efficiency of our method.
>
> 5) We have changed “three advantages” in the main text to “three characteristics”.
>
> 6) Figure 4 is a kernel density estimator that shows the density of the improvements of our model over the baseline. The figure illustrates that a majority of the improvements for few classes span over a large range of up to 40% improvement. The figure also illustrates that the decreases in performance in Medium and Many are constrained to within 10 points, a much smaller range.
> The alphas for a weak classifier do not sum to 1 after gamma regularization.
>
> References
> 1. Goodfellow, Ian, et al. “Machine Learning Basics: Capacity, Overﬁtting and Underﬁtting”. Deep learning. Vol. 1. No. 2. Cambridge: MIT press, 2016, pp. 116-120.

---

### Decision · Program_Chairs · 2021-01-07
**Final Decision**

**Decision:**

Reject

**Comment:**

The paper proposes to create models that address tail classes by computing a linear combination over models (concatenated weight vectors). Reviewers had grave concerns about the technical contribution, including justification of linear averaging of non-linear models, and about the experimental results, which improve on tail classes but hurt overall performance. As a result, the paper cannot be accepted to ICLR.